# Microwave Ablation of Liver, Kidney and Lung Lesions: One-Month Response and Manufacturer’s Charts’ Reliability in Clinical Practice

**DOI:** 10.3390/s22113973

**Published:** 2022-05-24

**Authors:** Julien Frandon, Philippe Akessoul, Tarek Kammoun, Djamel Dabli, Hélène de Forges, Jean-Paul Beregi, Joël Greffier

**Affiliations:** IMAGINE UR UM 103, Department of Medical Imaging, Nîmes University Hospital, Montpellier University, 30029 Nîmes, France; philippe.akessoul@chu-nimes.fr (P.A.); tarek.kammoun@chu-nimes.fr (T.K.); djamel.dabli@chu-nimes.fr (D.D.); helene.deforges@chu-nimes.fr (H.d.F.); jean.paul.beregi@chu-nimes.fr (J.-P.B.); joel.greffier@chu-nimes.fr (J.G.)

**Keywords:** interventional radiology, percutaneous ablation, tumour response, microwave, oncology

## Abstract

Microwave ablation systems allow for performing tumoral destruction in oncology. The objective of this study was to assess the early response and reliability of the microwave ablation zone size at one month for liver, kidney and lung lesions, as compared to the manufacturer’s charts. Patients who underwent microwave ablation with the Emprint^TM^ ablation system for liver, kidney and lung lesions between June 2016 and June 2018 were retrospectively reviewed. Local response and ablation zone size (major, *L*, and minor, *l*, axes) were evaluated on the one-month follow-up imaging. Results were compared to the manufacturers’ charts using the Bland–Altman analysis. Fifty-five patients (mean age 68 ± 11 years; 95 lesions) were included. The one-month complete response was 94%. Liver ablations showed a good agreement with subtle, smaller ablation zones (*L*: −2 ± 5.7 mm; *l*: −5.2 ± 5.6 mm). Kidney ablations showed a moderate agreement with larger ablations for L (*L*: 8.69 ± 7.94 mm; *l*: 0.36 ± 4.77 mm). Lung ablations showed a moderate agreement, with smaller ablations for l (*L*: −5.45 ± 4.5 mm; *l*: −9.32 ± 4.72 mm). With 94% of early complete responses, the system showed reliable ablations for liver lesions, but larger ablations for kidney lesions, and smaller for lung lesions.

## 1. Introduction

Thermoablation is now part of the wide therapeutic arsenal in oncology. Several studies have shown the non-inferiority of ablation techniques compared with conventional surgery in the treatment of some liver [1], pulmonary [2] or renal tumours [3]. These techniques led to a lower complication rate, a shorter hospital stay and an overall cost reduction. They are preferable for patients with important comorbidities or those who refuse surgery.

Ablation techniques include radiofrequency, microwave, cryotherapy, high-intensity focused ultrasound (HIFU) or laser. Microwave ablation is emerging as a favoured thermal technique thanks to technological advances allowing for a faster treatment time, a larger ablation zone and its use in various tissues as compared with the other ablation techniques [4]. The result is an enlarged ablation zone and theoretically fewer heat sink effects, such as cooling related to the proximity of a blood vessel [5,6]. Another advantage of microwave is the possibility to modulate the size and shape of the ablation zone by modifying the power delivered and the heating duration. To do this, the manufacturer proposes ex or in vivo indicative chart predictions to allow adaptation of the treatment to the proposed ablation with a given equipment (same generator and same needle).

There are various microwave ablation systems available, with different frequencies and technologies, thus producing different sized and shaped ablation zones [7]. In all systems, a difference between the expected volume provided by the manufacturer’s chart predictions and the effective ablation volumes is frequently reported [8]. The Emprint^TM^ Ablation system proposes an implemented microwave system with the Thermosphere^TM^ Technology, presented as more reliable and reproducible than others, and producing more spherical ablation zones [9].

The objective of the present study was to evaluate the early local response and reliability of the procedure in clinical practice, i.e., the size of the ablation at one month for kidney, liver and lung lesions compared with the chart provided by the manufacturer.

## 2. Patients and Methods

### 2.1. Patients

This monocentric study received approval from our Institutional review board for a retrospective review of consecutive patients who underwent microwave ablation for primary or secondary liver, kidney or lung tumours from June 2016 to June 2018. Patients with no reliable measurement of the ablation zone or with lesions treated with multiple overlapping ablations were not included.

### 2.2. Ablation Procedure

The ablations were conducted by two interventional radiologists with more than three years of experience in the technique. The microwave generator was the Emprint^TM^ system with Thermosphere^TM^ Technology from Medtronic^®^ (Minneapolis, MN, USA). Both the power delivered as well as the heating duration were at the operator’s discretion, depending on the charts provided by the manufacturer. In vivo charts were used for liver and lung lesions, and ex vivo charts for kidney lesions (Table 1). The choice of the imaging technique used during needle placement, ultrasound, conventional CT scan or cone beam CT depended on the tumour location and was left to the operator’s discretion.

### 2.3. Endpoints and Assessments

The primary endpoint was the local response on the first post-ablation CT scan or MR follow-up imaging scheduled at around one month for liver, kidney and lung lesions. Ballistic and size of the ablation zone were compared to the prior imaging to evaluate the complete tumour ablation. Secondary objectives were tumours’ characteristics, the size of the ablation zone and the comparison with the manufacturer’s chart. All measurements were performed by one observer (P.A.) on an axial acquisition at the portal time for the liver and kidney lesions, and with a parenchymal window for the lung lesions. The major (*L*) and minor (*l*) axes were measured for all ablation zones. A sphericity index (SI) was used [10,11], defined as the ratio between the volume of the ellipsoid and the volume of a sphere, whose diameter would be the largest axis of the ellipsoid. The ablation volume was calculated as the ellipsoid volume, as: $4\pi l^{2}L/3$. Using the major axis, the volume of the sphere was calculated as: $4\pi L^{3}/3$.

Thus, the sphericity index was calculated as follows: $$\mathrm{SI}=\frac{4\pi l^{2}L/3\;}{4\pi L^{3}/3}=\frac{l^{2}}{L^{2}}.$$

A sphericity index of 1.0 corresponded to a perfect sphere, and a sphericity index approaching 0 to an extreme ellipsoid.

### 2.4. Statistical Analysis

Data were reported as mean values with standard deviations and ranges. The values were compared using the Student’s *t*-test (parametric variables) or the Wilcoxon–Mann–Whitney test (nonparametric variables). Pearson correlation coefficients between the ablation zone measurements at one month (major and minor axes, sphericity index) and the manufacturer’s charts were assessed. One-month measurements and manufacturer’s measurements were compared using a Bland–Altman analysis, taking the manufacturer’s charts as a reference. A *p*-value < 0.05 was considered statistically significant.

## 3. Results

### 3.1. Patients and Lesions Characteristics

During the study period, 77 patients with 113 lesions (liver n = 73, kidney n = 18, lung n = 21 lesions) underwent microwave thermoablations using the Emprint^TM^ ablation system (Figure 1). Twelve (12/77, 16%) patients with eighteen lesions (18/113, 16%) were excluded due to multiple overlapping ablations for the same lesion (n = 15 lesions) or no reliable measurements (n = 3 lesions, one lost to follow-up, one bilioma, one patient with an ablation outside the range of the manufacturer’s charts).

Finally, 55 patients with a mean age of 68 ± 11 years (range: 33–91) with 95 lesions were included: 63 liver lesions (mean lesion long axis: 19.5 ± 7.6 mm), 14 kidney lesions (24.6 ± 8.7 mm) and 18 lung lesions (9.9 ± 5.5 mm) (Figure 2, Table 2). The indications were primary (n = 29/55, 53%) or secondary malignant tumours (n = 26/55, 47%), mainly metastases from colorectal carcinoma (n = 24/26, 92%) (Table 2). Among them, 33/55 (60%) patients had a single ablation and 22/55 (40%) patients had 2 to 4 simultaneous ablations.

### 3.2. Early Local Response

One month after the procedure, 89 ablations (89/95, 94%) were complete. Five liver ablations were partial, due to non-optimal needle placement during the ultrasound-guided treatment. These lesions were not clearly identified on sonography despite MRI fusion. One kidney ablation was partial, due to a tumour size of 31 mm with no optimal margins. All these patients received a second ablation treatment with CT guidance and needle placement guided by the prior ablation zone. This second treatment was not included in the study due to overlapping ablation on the same lesion.

### 3.3. Microwave Ablations’ Reliability

The ablation zones data are presented in Table 3, and 73% of the measurements were performed on MRI. The Bland–Altman analysis results for *L* and *l* are presented in Figure 3. Concerning liver lesions, 28 patients (28/31, 90%) with 58 ablations (58/63, 92%) underwent an MRI, and 3 patients (3/31, 10%) with 5 ablations (5/63, 8%) had a CT scan performed at a median time of 32 ± 7 days (range: 22–48) after treatment. The results of the Bland–Altman test showed a good agreement with a subtle, smaller, less spherical ablation zone than expected according to the manufacturer’s chart: −2 ± 5.7 mm for *L*, −5.2 ± 5.6 mm for *l*, −0.15 ± 0.18 for the SI. Correlations were moderate for *L* (r = 0.69, *p* < 0.001) and *l* (r = 0.59, *p* < 0.001). There was no correlation for SI (r = 0, *p* = 0.99).

For kidney lesions, 12 patients (12/13, 92%) with 13 ablations (13/14, 93%) had an MRI and 1 patient (1/13, 10%) with 1 ablation (1/14, 7%) had a CT scan performed at a median time of 32 ± 8 days (range: 12–50) after treatment. The Bland–Altman test showed a moderate agreement with a bigger and less spherical ablation zone than expected: +8.69 ± 7.94 mm for *L*, +0.36 ± 4.77 mm for l, −0.28 ± 0.27 for the SI (Figure 2). There were no significant correlations for *L* (r = 0.41, *p* = 0.14), *l* (r = 0.48, *p* = 0.08) or SI (r = −0.36, *p* = 0.21).

For lung lesions, 11 patients (11/11, 100%) with 18 ablations (18/18, 100%) had a CT scan performed at a median of 32 ± 6 days (range: 15–47) after treatment. A moderate agreement with a smaller, less spherical ablation zone than expected was found using the Bland–Altman test: −5.45 ± 4.5 mm for *L*, −9.32 ± 4.72 mm for *l*, −0.24 ± 0.2 for the SI (Figure 2). Correlations were strong for *L* (r = 0.76, *p* < 0.001) and *l* (r = 0.75, *p* < 0.001). There was no correlation for SI (r = 0.24, *p* = 0.33).

## 4. Discussion

The microwave ablation system evaluated in this study showed a very good early response, with 94% of complete ablation on the one-month follow-up imaging. The size of the ablation zone showed a good agreement for liver and moderate agreement for lung and kidney lesions with the ablation zone announced in the manufacturer’s chart. We reported smaller ablation zones for liver and lung lesions, and bigger ablation zones for kidney lesions. All ablations were less spherical than expected.

For hepatic lesions, the ablation area was significantly smaller than expected on the first follow-up imaging one month after the procedure, with only small differences for the major (2 mm) and minor axes (5 mm) but resulting in an ablation zone less spherical than expected. Although the differences were significant, the underestimation of 2 and 5 mm is acceptable in clinical practice. The early local response was very good, with 92% of complete ablations, concordant with that previously described in [10]. Smaller ablation zones at one month were expected because of tissue retraction and were different from previous studies that evaluated ablation areas on immediate postoperative imaging [11,12]. The follow-up was performed at one month according to the clinical practice in our centre. Thermosphere^TM^ was shown to perform reproducible and expected ablation zones at 100 Watts, but it may not reflect overall clinical practice, which is more heterogeneous, with the possibility of using the 25, 50, 75 and 100 Watts charts. In our study, we have not restricted power use to 100 Watts, it was left to the operator’s discretion as to reflect standard clinical practice. Although the sphericity index was lower than expected from the manufacturer’s charts (0.63 vs. 0.78), the Covidien^®^ material technology seems to produce more spherical ablation zones for liver lesions than other currently available devices such as Certus PR^®^ and AMICA^®^ materials, which have reported in vivo sphericity indices of 0.49 and 0.39, respectively [8].

For renal ablations, the major axis obtained on the first follow-up imaging was significantly longer than expected according to the manufacturer’s charts, with a potentially clinically significant difference (8.69 ± 7.94 mm) in case of proximity with structures at risk, such as the pyelon or ureter. There was no correlation with the chart. This result might be explained by the specific peritumoral environment. Indeed, percutaneous renal ablations often concerned cortical lesions located at the interface between the kidney and retroperitoneal fat. The surrounding fat has dielectric properties, reflecting the electromagnetic energy that may impact on the ablation [13]. A previous study showed that depending on tissue limitations, the shape of the thermoablation zone may range from 7.5% to 23.4% without any effect on the minor axis [14]. Another explanation may be the lack of an in vivo chart for renal ablations, unlike for liver and lung. A study evaluating a high-frequency microwave on ex vivo bovine liver and in vivo pig liver tissue showed that whatever the power delivered, the ablation zones developed differently according to their in vivo or ex vivo environment [15]. These differences, with larger ablation zones than expected, should be taken into consideration in clinical practice.

For pulmonary ablations, the major and minor axes obtained, and thus the ablation zone, were significantly much smaller than expected according to the manufacturer’s charts. Indeed, ablations performed were of 3 cm. For bigger lesions, the underestimation of the small axis could be a problem, with only partial ablation obtained. However, the correlation with the chart was strong, probably due to the use of in vivo references and to the homogeneous environment of lungs. Treated lesions were small (around 1 cm), with 100% of complete response. The ablations were less spherical than expected. This might, as suggested for liver lesions, be explained by the delay of evaluation, one month after the procedure. Kodama et al. showed, in a swine model, that the best temporal evaluation for microwave ablation was one week after the procedure [16]. Performing an earlier follow-up imaging, at one week, for example, may allow a better evaluation of the ablation zone.

The significant difference found on the ablation shape with a sphericity index lower than expected might be due to tissue retraction. Microwaves, with high temperatures generated, induce tissue shrinkage through dehydration and collapse. An ex vivo study showed that microwaves produced a more important tissue retraction (about 30%) than radiofrequency (15%) in the liver [17]. The authors showed that this phenomenon has a greater impact on the minor than on the major axis. In the same study, retraction was shown to be even more important in the lungs (about 50%), but with no difference between microwave and radiofrequency [17]. This was coherent with our results, with a smaller minor axis and a lower sphericity index for pulmonary ablations than for liver ablations (0.52 for lung lesions, 0.63 for liver lesions). The phenomenon of retraction was also demonstrated in ex vivo kidney tissue in another study [18]. Additionally, Lee et al. confirmed this tissue-retraction effect in vivo for liver lesions in 65 patients. In their study, the ablation zone was evaluated earlier than in our study, with MRI follow-up 24 h after the procedure. They showed a shrinkage of the ablation zone greater for microwave than for radiofrequency ablations (−2.45 ± 0.47 mm for microwaves vs. 0.94 ± 0.38 mm for radiofrequency) [19].

This study had several limitations, mainly due to its retrospective and monocentric design. Imaging follow-up was performed at one month, according to the standard clinical practice in our Institute, but it may have resulted in an underestimation of the ablation size due to tissue retraction. An immediate post-procedural imaging using contrast-enhanced ultrasound [20] or CT perfusion imaging [21,22] could have been performed to estimate the real ablation volume, with less retraction effect. Additionally, only axial imaging was used to evaluate the major and minor axes. Indeed, the great majority of follow-up imaging was performed with MRI with standard axial acquisitions, without 3D acquisitions. We thus preferred to use an axial acquisition for more reproducibility between different imaging modalities. Nevertheless, these preliminary results could be used as a basis for future prospective and multicentric studies.

## 5. Conclusions

The microwave ablation system evaluated in this study showed a very good early local response, with 94% of complete ablations on the one-month follow-up imaging. For liver lesions, in comparison with the manufacturer’s charts, despite statistically significant differences, ablations were reliable. For kidney lesions, ablations were bigger than those expected in the charts, which should be taken into account by the operators. For lung lesions, the retraction effect was significant, and the ablation zone should be evaluated earlier than at one month.

## Figures and Tables

**Figure 1 sensors-22-03973-f001:**
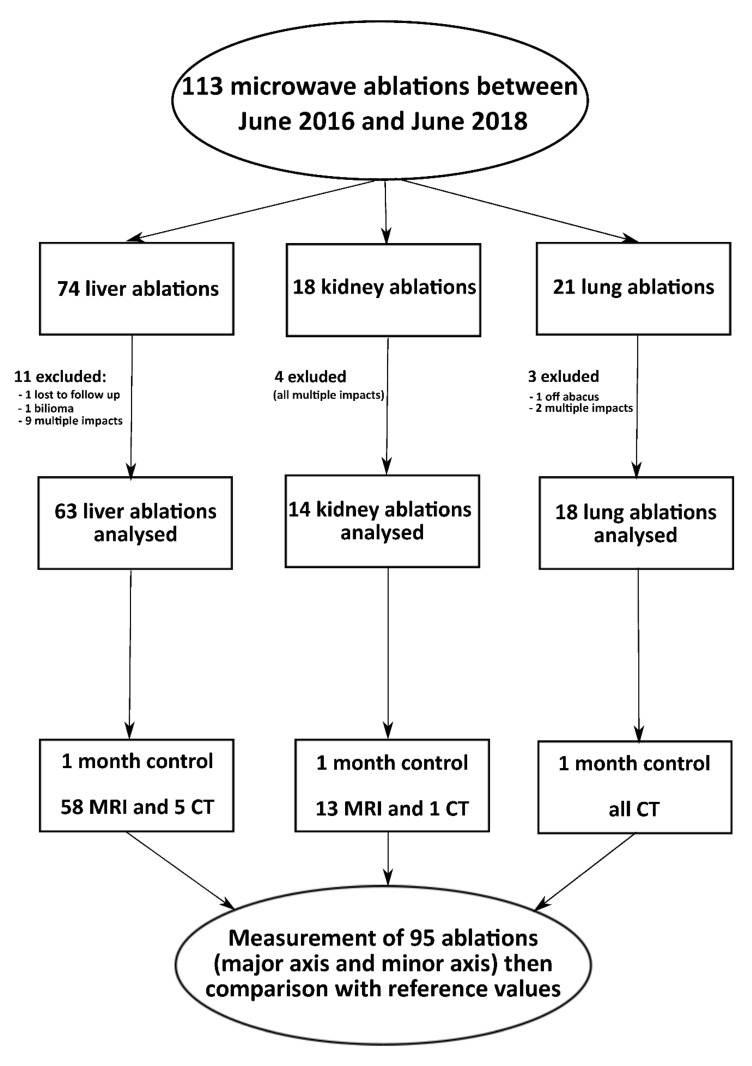
Flowchart of the study.

**Figure 2 sensors-22-03973-f002:**
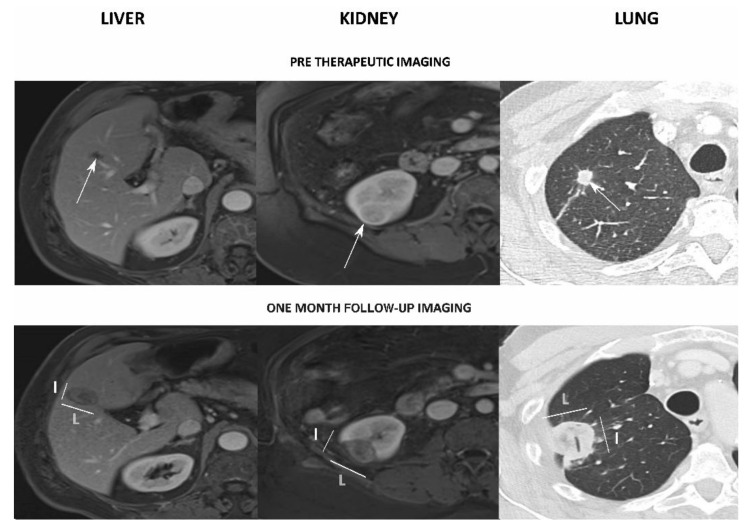
Examples of assessment of the ablation zones for liver (**left**), kidney (**middle**) and lung (**right**) lesions. **Top**: pre-treatment assessment of the target lesion (white arrows). **Bottom**: assessment on the one-month follow-up imaging, with the major (*L*) and minor (*l*) axes measurements. Note: air was found inside the ablation zone (tubular structure).

**Figure 3 sensors-22-03973-f003:**
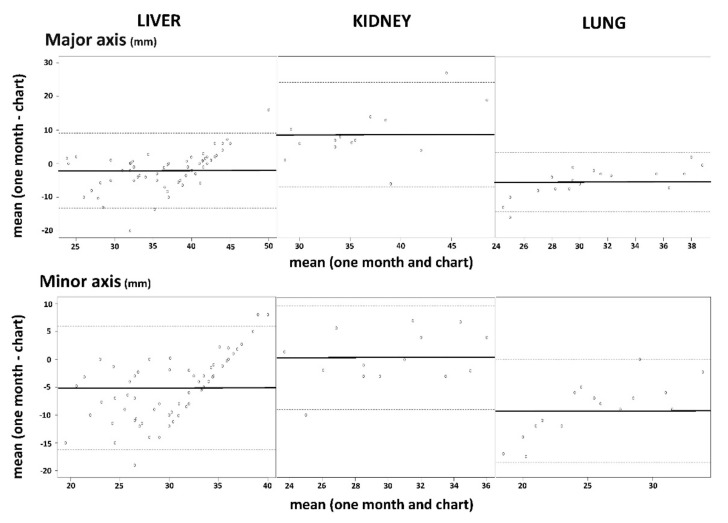
Bland–Altman plots showing agreement between the one-month evaluation and charts’ data for the major (**top**) and minor (**bottom**) axes, for liver (**left**), kidney (**middle**) and lung (**right**) lesions. Charts’ data were chosen as the reference. Note: Mean (one month–chart): mean difference between *L* and *l* on the charts and the ablation zone measured on the one-month follow-up patient imaging. Mean (one month and chart): mean of the sizes proposed by the charts and the ablation zone measured on the one-month follow-up patient image.

**Table 1 sensors-22-03973-t001:** Charts provided by the manufacturer for liver, kidney and lung ablations.

Organ	Power (Watt)/Time (Min) (Number of Ablations)	Major Axis Expected (mm)	Minor Axis Expected (mm)
Liver * (n = 63)	100 w/10 min (n = 23)	41	36
	100 w/7 min (n = 8)	40	35
	100 w/5 min (n = 1)	39	34
	100 w/4 min (n = 1)	38	33
	75 w/10 min (n = 7)	36	33
	75 w/7 min (n = 2)	35	32
	75 w/5 min (n = 5)	33	30
	75 w/4 min (n = 2)	32	29
	75 w/2.30 min (n = 8)	30	27
	45 w/5 min (n = 2)	27	27
	45 w/4 min (n = 2)	26	26
	45 w/2.30 min (n = 2)	24	23
Kidney ** (n = 14)	100 w/2.30 min (n = 2)	29	26
	75 w/7 min (n = 5)	32	31
	75 w/5 min (n = 2)	30	29
	75 w/4 min (n = 2)	28	27
	75 w/2.30 min (n = 2)	25	23
	45 w/5 min (n = 1)	25	25
Lung * (n = 18)	75 w/5 min (n = 1)	38	34
	75 w/4 min (n = 1)	38	33
	75 w/2.30 min (n = 3)	37	32
	45 w/10 min (n = 1)	34	30
	45 w/5 min (n = 4)	33	29
	45 w/2.30 min (n = 8)	32	27

*: in vivo chart. **: ex vivo chart.

**Table 2 sensors-22-03973-t002:** Patients’ characteristics.

Characteristics	Liver	Kidney	Lung
Patients, n (men/women)	31 (22/9)	13 (9/4)	11 (6/5)
Lesions, n	63	14	18
Mean age, years (SD)	68 (±10)	71 (±13)	65 (±9)
Tumour origin, n (%)			
Primitive	14/31 (45%)	12/13 (92%)	3/11 (27%)
Metastatic	17/31 (55%)	1/13 (8%)	8/11 (63%)
Metastatic origin, n (%)			
Colon	16/17 (94%)	1/1 (100%)	7/8 (87%)
Others	1/17 (6%)	0/1 (0%)	1/8 (23%)
Tumour size, mean (SD) [range], mm			
Long axis	19.5 (±7.6) [8;35]	23.7 (±6.5) [13;37]	9.9 (±5.5) [5;25]
Minor axis	15.5 (±6.2) [6;29]	19.4 (±6.6) [9;30]	7.5 (±4.8) [2;19]
Needle guidance (per lesion), n (%)			
CT	19/63 (30%)	12/14 (86%)	12/18 (67%)
CBCT	2/63 (3%)	2/14 (4%)	6/18 (33%)
US	42/63 (67%)	0/14 (0%)	0/18 (0%)
Follow-up imaging (per patient), n (%)			
MRI	28/31 (90%)	12/13 (92%)	0/11 (0%)
CT	3/31 (10%)	1/13 (8%)	11/11 (100%)
One-month response (per lesion), n (%)			
Complete	58/63 (92%)	13/14 (93%)	18/18 (100%)
Partial	5/63 (8%)	1/14 (7%)	0/18 (0%)

**Table 3 sensors-22-03973-t003:** Ablation zones sizes and sphericity index expected according to the manufacturer’s charts and measured on the one-month follow-up.

	Charts (mm)	One-Month Follow-Up (mm)	*p*-Values
**Liver**			
Major axis, mean (SD) [range] (mm)	37.4 (±5.1) [24;44]	35.3 (±7.9) [22–46]	**0.006**
Minor axis, mean (SD) [range] (mm)	32.9 (±3.9) [23;36]	27.7 (±7.0) [12–38]	**<0.001**
Sphericity index, mean (SD) [range]	0.78 (±0.06) [0.7;1]	0.63 (±0.17) [0.32–0.9]	**<0.001**
**Kidney**			
Major axis, mean (SD) [range] (mm)	32.0 (±5.1) [24;42]	40.7 (±8.5) [29–58]	**0.003**
Minor axis, mean (SD) [range] (mm)	29.9 (±3.7) [23;36]	30.3 (±5.3) [20–38]	0.67
Sphericity index, mean (SD) [range]	0.89 (±0.09) [0.7;1]	0.60 (±0.23) [0.43–0.97]	**0.005**
**Lung**			
Major axis, mean (SD) [range] (mm)	33.7 (±3.3) [30;40]	28.2 (±6.4) [17–39]	**<0.001**
Minor axis, mean (SD) [range] (mm)	29.5 (±3.0) [27;36]	20.2 (±6.6) [10–33]	**<0.001**
Sphericity index, mean (SD) [range]	0.77 (±0.03) [0.71;0.81]	0.52 (±0.2) [0.25–0.93]	**<0.001**

Sizes according to the manufacturer’s charts were elaborated in accordance with the treatments administered to the patients. Figures in bold indicate significant differences (<0.05).

## Data Availability

The data presented in this study are available upon request from the corresponding author.

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
