# Peer review of "Microwave Ablation of Liver, Kidney and Lung Lesions: One-Month Response and Manufacturer’s Charts’ Reliability in Clinical Practice"

_sensors, 2022, doi:10.3390/s22113973_

Round 1

Reviewer 1 Report

The paper deals with retrospective study evaluating differences between manufacturer predictions of ablation size and actually measured ablation zone dimensions from 1 month follow-up images. I find this study and its findings important to be published. Please refer to following minor points on how to update paper.

Abstract: typo: Please change term to "Microwave ablation systems".

Introduction, 2nd paragraph: The statement that "Microwave ablation does not depend on electrical conductivity [5]" is suspicious. As frequency increases, the name of term may change to effective conductivity, but still represent EM losses within tissue. Furthermore, Effective conductivity plays major role in shaping and extent of direct heating zone with electromagnetic heating losses. Therefore, I recommend deleting this statement.

Lines 70-72: Please check grammar of the sentence.

Figure 2, bottom right subplot in lungs:  Can authors comment on dark tubular structure in the middle of ablated region? It almost seems like insertion track, although it could be airway as well. If it is indeed insertion track, then labels for L, and l need to be changed.

Thank you
